# Epoxide-functionalization of polyethyleneimine for synthesis of stable carbon dioxide adsorbent in temperature swing adsorption

Woosung Choi[1,*], Kyungmin Min[1,*], Chaehoon Kim[1], Young Soo Ko[2], Jae Wan Jeon[2], Hwimin Seo[3], Yong-Ki Park[3] & Minkee Choi[1]

Amine-containing adsorbents have been extensively investigated for post-combustion carbon dioxide capture due to their ability to chemisorb low-concentration carbon dioxide from a wet flue gas. However, earlier studies have focused primarily on the carbon dioxide uptake of adsorbents, and have not demonstrated effective adsorbent regeneration and long-term stability under such conditions. Here, we report the versatile and scalable synthesis of a functionalized-polyethyleneimine (PEI)/silica adsorbent which simultaneously exhibits a large working capacity ($2.2\,mmol\,g^{-1}$) and long-term stability in a practical temperature swing adsorption process (regeneration under 100% carbon dioxide at 120 °C), enabling the separation of concentrated carbon dioxide. We demonstrate that the functionalization of PEI with 1,2-epoxybutane reduces the heat of adsorption and facilitates carbon dioxide desorption (>99%) during regeneration compared with unmodified PEI (76%). Moreover, the functionalization significantly improves long-term adsorbent stability over repeated temperature swing adsorption cycles due to the suppression of urea formation and oxidative amine degradation.

[1] Department of Chemical and Biomolecular Engineering, Korea Advanced Institute of Science and Technology, Daejeon 305–701, Republic of Korea. [2] Department of Chemical Engineering, Kongju National University, Cheonan 331–717, Republic of Korea. [3] Center for Convergent Chemical Process, Korea Research Institute of Chemical Technology, Daejeon 305–600, Republic of Korea. * These authors contributed equally to this work. Correspondence and requests for materials should be addressed to M.C. (email: mkchoi@kaist.ac.kr).

Global warming due to $CO_2$ emissions is an urgent issue because it can cause severe climate change and environmental catastrophe[1]. The $CO_2$ emissions associated with human activities are mainly due to the use of fossil fuels, and the implementation of $CO_2$ capture and storage technologies in power plants has been proposed as a means of enabling the continued use of fossil fuels in the short term[2,3]. In particular, post-combustion $CO_2$ capture has attracted a lot of attention because the technology can be easily retrofitted into the existing power plants[2,3]. Among various $CO_2$ capture methods, $CO_2$ capture using aqueous alkanolamine solutions has been intensively researched and used industrially for over 50 years[4]. However, the amine scrubbing processes involve the intrinsic limitations of potential environmental and health concerns due to volatile amine loss, corrosion problems and high energy consumption for solvent regeneration[2,5]. To overcome these limitations, solid adsorbents have emerged as a promising alternative by virtue of their non-corrosive property and lower energy penalty for regeneration[2,6,7].

Various solid adsorbents such as amine-functionalized porous materials[8–29], zeolites[30–33], carbons[34,35] and metal organic frameworks[36–41] have been widely investigated for $CO_2$ capture. Among these adsorbents, amine-functionalized porous materials have been most extensively investigated due to their effective ability to chemisorb low-concentration $CO_2$ (<15%) from a wet flue gas[2,6,7]. The adsorbents can be prepared by impregnating polymeric amines such as polyethyleneimine (PEI) into porous supports[8–22], grafting aminosilanes on the pore surface[21–26] and in situ polymerization of amine monomers within the support pores[26,27]. By introducing high-loading amines into large-porosity supports, adsorbents with large $CO_2$ uptakes (generally above 2 mmol g$^{-1}$) could be readily prepared. However, as critically mentioned in several papers[6,7,22,28], most previous studies have primarily focused on increasing the $CO_2$ uptake of adsorbents during the adsorption step, and there have been very few reports describing practically meaningful regeneration methods for the adsorbents. In earlier works, adsorbents were almost always regenerated by increasing temperature under an inert gas purge (for example, $N_2$)[8,10,12–15,17–21,23,26,27], which ultimately does not result in the separation of $CO_2$. Therefore, several critical missing links remain in the development of a practical $CO_2$ capture process, specifically, the demonstration of energy-efficient adsorbent regeneration enabling the purified $CO_2$ separation and long-term material stabilities under such conditions. Another important issue is the cost and scalability of adsorbent synthesis, which has often been ignored in academic studies. Considering that material syntheses requiring the use of corrosive and expensive chemicals and multi-step procedures are difficult to scale up, an enormous gap appears to exist between many of the current academic studies and practical adsorbent development.

The regeneration of solid adsorbents can be carried out using pressure swing adsorption (PSA), vacuum swing adsorption (VSA) and temperature swing adsorption (TSA) processes. However, PSA and VSA require a large amount of energy in compressing or applying a vacuum to large volumes of low-pressure flue gas stream[6,7,37,42,43]. In addition, the availability of waste heat from a power plant as a source of regeneration energy makes TSA more attractive than PSA and VSA[37,42]. However, to design a practical TSA process which enables the separation of highly concentrated $CO_2$, the $CO_2$ should be desorbed from the adsorbents under a $CO_2$-rich atmosphere using a primarily thermal driving force (that is, without an inert gas purge; the separated $CO_2$ can be recycled as a purge gas). Unfortunately, in the case of amine-functionalized adsorbents, a relatively high temperature (>120 °C) must be used

to desorb $CO_2$ under a $CO_2$-rich atmosphere, where a significant deactivation of amines via urea formation (that is, dehydrative condensation between amines and $CO_2$) takes place[16–24]. Recently, Jones and colleagues[7,28] proposed the use of high-temperature steam for regenerating the amine-functionalized adsorbents, because it can suppress the urea formation according to Le Chatelier's principle and reduce the partial pressure of $CO_2$ under the desorption condition. The added steam can be removed as liquid water after condensation. For this process to be feasible, an economic source of steam needs to be available and the adsorbents should have high hydrothermal stability[7,15,29].

In the present work, we report a highly versatile and scalable synthesis of functionalized-PEI/silica composite adsorbent that simultaneously exhibits a large $CO_2$ working capacity (2.2 mmol g$^{-1}$) as well as long-term stability in a TSA condition (adsorbent regeneration under 100% $CO_2$ at 120 °C) even without the addition of steam. The use of 100% $CO_2$ as a purge gas enables the separation of $CO_2$ in a highly concentrated state, but

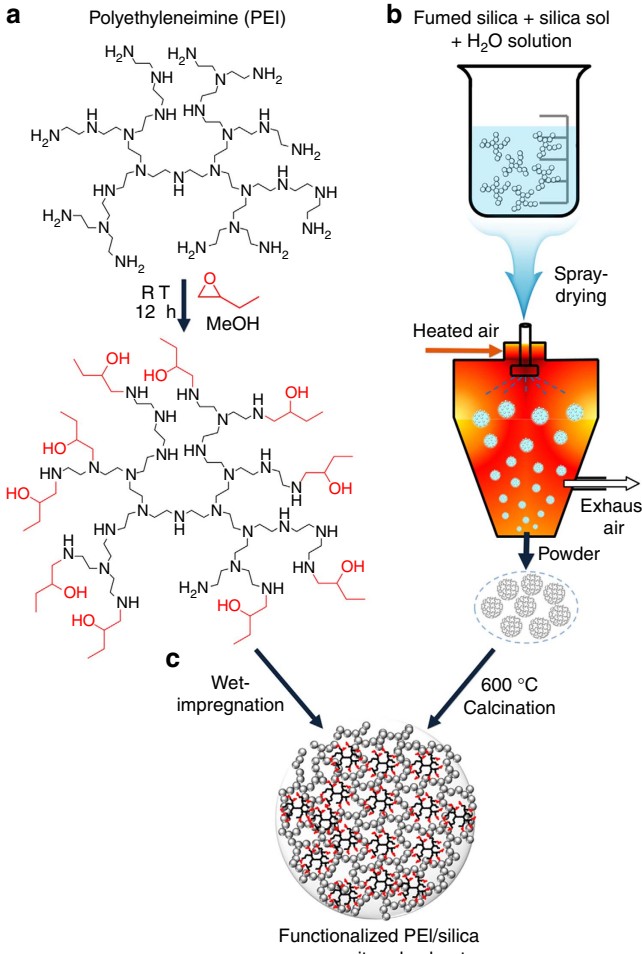

**Figure 1 | Schematic representation of the scalable synthesis of the $CO_2$ adsorbent.** (**a**) Functionalization of PEI with 1,2-epoxybutane was carried out by a single-step addition reaction. To control the degree of functionalization, varied amounts of 1,2-epoxybutane were added to 17 wt% methanolic solution of PEI and the reaction was carried out at room temperature for 12 h. (**b**) Silica microspheres having extra-large porosity (1.7 cm$^3$ g$^{-1}$) were synthesized by spray-drying a water slurry containing 10 wt% fumed silica and 0.5 wt% silica sol as a binder, followed by air calcination at 600 °C to sinter the fumed silica into a 3D porous structure. (**c**) The methanolic solutions of the functionalized PEIs obtained in **a** were impregnated into the pre-made silica microspheres.

at the same time it is the most unfavourable atmosphere for adsorbent regeneration in terms of $CO_2$ partial pressure and amine deactivation via urea formation. In this work, it will be demonstrated that the controlled functionalization of PEI with 1,2-epoxybutane can enhance $CO_2$ desorption under the TSA cycle by reducing the heat of $CO_2$ adsorption. Furthermore, the functionalization significantly improves the adsorbent stability over the repeated TSA cycles due to the suppression of urea formation and oxidative amine degradation.

## Results

**Functionalization of PEI and adsorbent preparation**. Functionalization of PEI with 1,2-epoxybutane was carried out by a simple single-step addition reaction at ambient conditions (Fig. 1a). To control the degree of functionalization, varied amounts of 1,2-epoxybutane were added dropwise to 17 wt% methanolic solution of PEI (MW 1,200, nitrogen content: 22 $mmol_N g^{-1}$) under stirring. The reaction was carried out at room temperature for 12 h. The functionalized PEIs were denoted as $n$EB-PEI, where $n$ indicates the molar ratio between 1,2-epoxybutane and the nitrogen content within PEI initially used for the reaction. Elemental analysis after the vacuum evaporation of methanol and possibly unreacted 1,2-epoxybutane at 60 °C for 12 h (Table 1) showed that the O/N elemental ratios of the resultant $n$EB-PEIs are very close to those predicted from the initial reactants' stoichiometry ($n$). The result confirms that the addition reaction goes almost completely. This can be attributed to the high ring-opening reactivity of epoxides in the presence of nucleophilic amines[44].

Liquid-phase $^{13}C$ NMR (Fig. 2a, Supplementary Fig. 1 and Supplementary Note 1) was used to characterize the amine state distributions in the PEI and $n$EB-PEIs[45,46]. According to the quantitative analysis, the unmodified PEI initially possessed the primary (1°):secondary (2°):tertiary (3°) amine ratio of 36:37:27, respectively (Fig. 2b). After the PEI functionalization with 1,2-epoxybutane, the portion of 1° amine gradually decreased, while the portions of 2° and 3° amines increased. It should be noted that the increase in 2° amine portion was much faster than the increase of 3° amine portion. This indicates that the functionalization quite selectively converted 1° amines to 2° amines, while the alkylation of 2° amines to 3° amines was relatively suppressed. For instance, in the case of 0.37EB-PEI, ca. 80% 1,2-epoxybutane appears to have been used for the conversion of 1° amines to 2° amines, while the other 20% was used for the conversion of 2° amines to 3° amines. Such a preferential alkylation of 1° amines to 2° amines is highly desirable in order not to sacrifice the $CO_2$ adsorption capacities, because 3° amines capture $CO_2$ much less efficiently than 1° and 2° amines[2,6,7].

As a solid support for the PEI and $n$EB-PEIs, a porous silica was synthesized by spray-drying a water slurry containing 10 wt%

fumed silica and 0.5 wt% silica sol as a binder (Fig. 1b), followed by air calcination at 600 °C to sinter the fractal-like fumed silica particles into a three-dimensional (3D) porous network (Fig. 2d). The silica spheres have the particle sizes mainly in the range of 75–200 μm (Fig. 2c and Supplementary Fig. 2), which is suitable for fluidized bed operations in a large-scale $CO_2$ capture process. The porous silica microspheres showed a slightly smaller BET surface area (299 $m^2 g^{-1}$) but significantly larger pore volume (1.7 $cm^3 g^{-1}$) than the original fumed silica powder (300 $m^2 g^{-1}$ and 1.0 $cm^3 g^{-1}$) having only interparticular porosity (Fig. 2e and Table 1). The silica showed a pore size distribution in the range of 20–100 nm (Fig. 2e). We confirmed that the silica microspheres impregnated with PEI showed $CO_2$ adsorption capacities and kinetics superior even to the adsorbents prepared with ordered mesoporous silicas such as MCM-41 (ref. 47) (pore volume: 1.0 $cm^3 g^{-1}$, Supplementary Fig. 3) and SBA-15 (ref. 48) (1.1 $cm^3 g^{-1}$) at the same 50 wt% PEI loading (Fig. 2f). The result indicates that the ordered arrangement of silica mesoporosity derived from supramolecular templating[47,48] is not essential for achieving good $CO_2$ adsorption performance, although many of the earlier studies used this type of silica support[8–10,12,19–22]. Recently, several groups have reported similar results showing that mesopore order is not important for designing high-performance $CO_2$ adsorbents if a silica support possesses sufficiently large 3D porosity that enables efficient $CO_2$ diffusion[11,18]. We also compared long-term steam stabilities of the composite adsorbents made from MCM-41, SBA-15 and the present silica microspheres. After the treatment under 100% steam at 120 °C for 7 days, the composite adsorbents were calcined at 600 °C and the pore structures of the silica supports were analysed using $N_2$ adsorption–desorption at −196 °C (Supplementary Fig. 4). The results showed that the $N_2$ adsorption–desorption isotherm of the steam-treated silica microsphere changed only slightly (<5% pore volume loss) compared with that of a pristine silica microsphere. In clear contrast, the isotherms of SBA-15 and MCM-41 significantly changed after the steam treatment compared with those of pristine silicas. In all, 39 and 46% decreases in pore volume were observed for SBA-15 and MCM-41, respectively. The present results clearly indicate that the present silica microspheres possess much higher hydrothermal stability compared with ordered mesoporous silicas such as SBA-15 and MCM-41. The remarkably higher stability of the silica microspheres can be attributed to their significantly larger silica framework thickness (10–15 nm) than those of SBA-15 (ca. 3 nm) (ref. 49) and MCM-41 (ca. 1 nm) (ref. 50). The present synthesis of silica microspheres having large porosity and hydrothermal stability does not require the use of expensive and corrosive chemicals (for example, surfactant and acid/base), and thus it is cost-efficient and environmentally friendly.

For the preparation of the polymer–silica composite $CO_2$ adsorbents, the methanolic solutions of PEI and $n$EB-PEIs were

**Table 1 | Physical characteristics of the prepared materials.**

| Sample | O/N (exp)* | O/N (theor) | Amines (%)† | | | $S_{BET}$ ($m^2 g^{-1}$)‡ | $V_{total}$ ($cm^3 g^{-1}$)§ | N contents ($mmol_N g^{-1}$) |
|---|---|---|---|---|---|---|---|---|
| | | | 1° | 2° | 3° | | | |
| $SiO_2$ | — | — | — | — | — | 299 | 1.7 | — |
| PEI/$SiO_2$ | — | — | 36 | 37 | 27 | 37 | 0.34 | 10.4 |
| 0.15EB-PEI/$SiO_2$ | 0.15 | 0.14 | 23 | 44 | 33 | 40 | 0.37 | 8.9 |
| 0.37EB-PEI/$SiO_2$ | 0.37 | 0.36 | 10 | 56 | 34 | 43 | 0.37 | 6.7 |
| 0.54EB-PEI/$SiO_2$ | 0.54 | 0.57 | 2 | 61 | 37 | 45 | 0.38 | 5.9 |

* O/N mole ratio was determined from elemental analysis.
† Amine state distributions of PEI and functionalized PEIs were calculated from $^{13}C$ NMR analysis.
‡ BET surface areas were determined in the $P/P_0$ range of 0.05–0.20.
§ Total pore volumes ($V_{total}$) were evaluated at $P/P_0 = 0.99$.

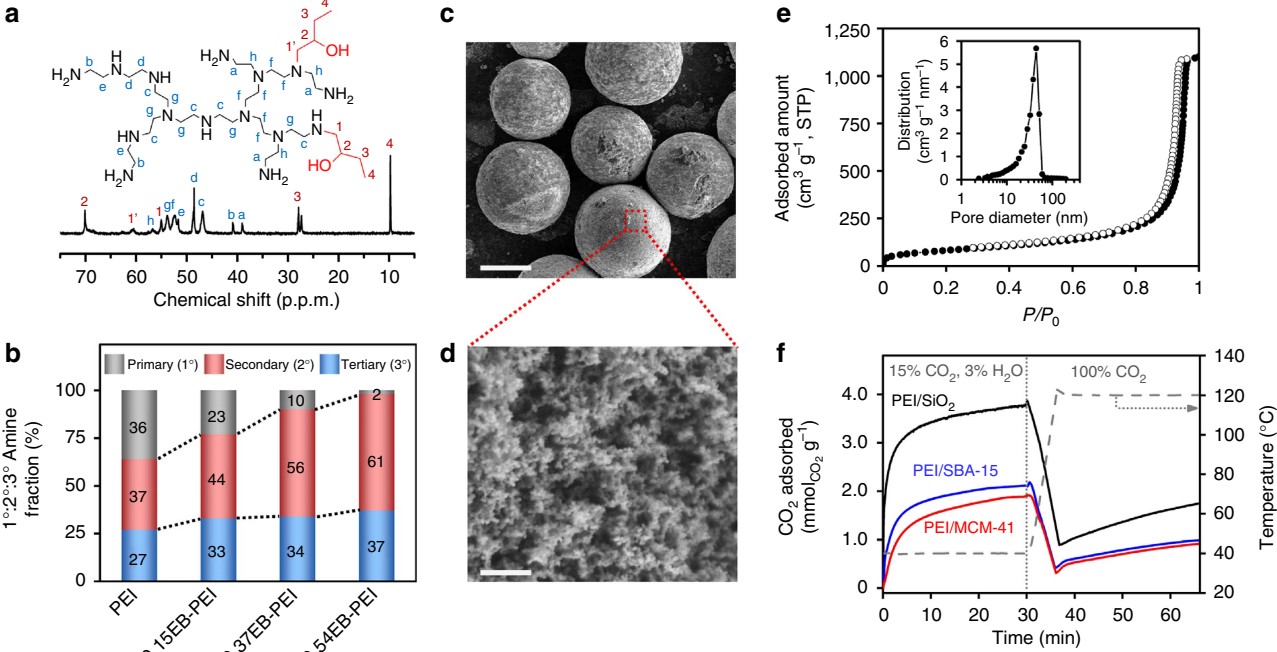

**Figure 2 | Characterization of functionalized PEIs and a microsphere silica support. (a)** Molecular structure and a representative liquid-phase $^{13}C$ NMR spectrum for a functionalized PEI (0.37EB-PEI). **(b)** Amine state distributions of PEI and functionalized PEIs ($n$EB-PEI) analysed by $^{13}C$ NMR. Quantitative NMR analysis of the amine state distribution was carried out using the following equation; primary(1°):secondary(2°):tertiary(3°) = $(A_a + A_b)$:
$(A_c + A_d + A_e + A_1)/2$:$(A_f + A_g + A_h + A_{1'})/3$, where $A_i$ is the integrated peak area for $i$ species[44,45]. **(c,d)** Scanning electron microscope image of the silica microspheres synthesized by a spray-drying of a fumed silica. Scale bars, 50 µm **(c)**, 200 nm **(d)**. **(e)** $N_2$ adsorption–desorption isotherm of the silica microspheres measured at −196 °C (inset: corresponding pore size distribution obtained using the Barrett − Joyner − Halenda (BJH) equation).
**(f)** Comparison of $CO_2$ adsorption–desorption profiles of various silicas impregnated with 50 wt% PEI in a TSA cycle (adsorption: 15% $CO_2$, 3% $H_2O$, 2% Ar in $N_2$ balance at 40 °C; desorption: 100% $CO_2$ at 120 °C). PEI impregnated in the silica microspheres (PEI/SiO$_2$) showed much higher $CO_2$ uptake than PEIs in ordered mesoporous silicas such as MCM-41 and SBA-15.

impregnated into the pre-made porous silica microspheres (Fig. 1c). The nominal polymer loading was fixed as 50 wt% of the composite adsorbents. In the case of $n$EB-PEIs, the methanolic solutions obtained after the functionalization reaction were directly used for impregnation, without additional purification steps, because the reaction goes to near a complete level as mentioned above. The physical properties of the prepared polymer–silica composite adsorbents are summarized in Table 1. All the adsorbents showed similar pore volumes in the range of 0.34–0.38 cm$^3$ g$^{-1}$. As the degree of functionalization ($n$) increased, the N content in the composite adsorbent decreased. This is reasonable because the loaded polymer weight was fixed as 50 wt%, while the molecular weight of the polymer was increased after the functionalization.

**CO$_2$ adsorption–desorption behaviour under a TSA cycle**. The CO$_2$ adsorption–desorption behaviour of the adsorbents was investigated under a TSA cycle. To simulate a practically meaningful TSA cycle, a wet flue gas (15% CO$_2$, 3% H$_2$O, 2% Ar in N$_2$ balance) was used for CO$_2$ adsorption at 40 °C, and a dry 100% CO$_2$ atmosphere was used for the adsorbent regeneration at 120 °C (Fig. 3a). CO$_2$ adsorption–desorption profiles were measured using a thermogravimetric analyser combined with a mass spectrometer (TGA-MS; Supplementary Fig. 5)[25]. To confirm the reliability of the measurement system, the CO$_2$ adsorption capacities were also cross-checked using a different breakthrough experimental setup (Supplementary Fig. 6), in which the CO$_2$ concentration was detected by a thermal conductivity detector after removing moisture with a −10 °C cold trap. In the measurements, all samples were diluted 10 times

by using sand (quartz) as a diluent for avoiding heat-transfer limitation (Supplementary Fig. 7). Both experimental setups gave consistent CO$_2$ adsorption amounts within an error range of 10% (Supplementary Table 1), which confirmed the reliability of the measurements.

The adsorbent prepared with an unmodified PEI (PEI/SiO$_2$) showed the largest CO$_2$ uptake of 3.8 mmol g$^{-1}$ in the adsorption step (Fig. 3a,b). This can be attributed to the largest nitrogen content of PEI/SiO$_2$ among the adsorbents (Table 1). In the regeneration step, the sample showed 76% desorption of the initially adsorbed CO$_2$ and thus the working capacity ('desorbable' CO$_2$ uptake) was determined to be 2.9 mmol g$^{-1}$. In the cases of adsorbents with the functionalized PEIs ($n$EB-PEI/SiO$_2$), CO$_2$ uptake gradually decreased with an increasing degree of functionalization ($n$) (Fig. 3a,b). This can be attributed to the reduced N contents (Table 1) as well as the slightly increased 3° amine fractions after the functionalization (Fig. 2b). However, it is noteworthy that more efficient CO$_2$ desorption was observed as the degree of functionalization ($n$) increased (Fig. 3a,b). As a result, the highly functionalized 0.37EB-PEI/SiO$_2$ and 0.54EB-PEI/SiO$_2$ samples showed complete desorption (>99%) of the initially adsorbed CO$_2$. Although PEI/SiO$_2$ showed a larger CO$_2$ uptake than $n$EB-PEI/SiO$_2$ samples during the adsorption step, the differences in working capacities were relatively smaller due to the more efficient desorption of CO$_2$ in the $n$EB-PEI/SiO$_2$ samples. The enhanced desorption of CO$_2$ after the functionalization can be attributed to the reduced heat of CO$_2$ adsorption (line plot in Fig. 3b), which can be explained by the fact that the functionalization resulted in the alkylation of amines with 2-ethyl-hydroxyethyl groups (–CH$_2$CH(C$_2$H$_5$)OH). The hydroxyethyl groups are

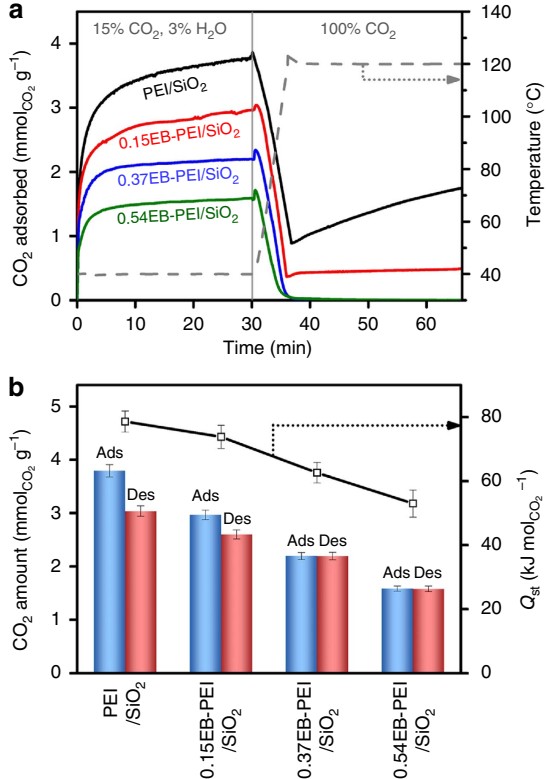

**Figure 3 | CO₂ adsorption–desorption behaviour of the adsorbents.**
(**a**) $CO_2$ adsorption–desorption profiles of the adsorbents in a TSA cycle
(adsorption: 15% $CO_2$, 3% $H_2O$, 2% Ar in $N_2$ balance at 40 °C; desorption:
100% $CO_2$ at 120 °C). (**b**) $CO_2$ adsorption/desorption amounts during the
TSA cycle (bars) and the heat of $CO_2$ adsorption (line plot) for the
adsorbents. The measurements were repeated three times and averaged.

well-known electron-withdrawing groups that can lower the
basicity of the amine centre, which can result in the weakened
interaction with $CO_2$ (ref. 51). Besides, the side-chain ethyl
groups can also increase the steric hindrance near the amine
centres and thus destabilize the carbamate species formed after
$CO_2$ adsorption[51]. Indeed, lowing the basicity and increasing the
steric hindrance of the amine centres have been used as major
strategies to reduce the energy penalty required for solvent
regeneration in amine scrubbing process[2].

For quantitative analysis of $CO_2$ adsorption kinetics, $CO_2$
uptake profiles measured with the breakthrough experimental
setup were fitted with the Avrami kinetic equation[52]
(Supplementary Fig. 8 and Supplementary Table 2):

$$Q_t = Q_e[1 - \exp((-k_a t)^{n_a})] \qquad (1)$$

where, $k_a$ (min$^{-1}$) is a rate constant, $n_a$ is the kinetic order of the
Avrami model, and $Q_t$ (mmol g$^{-1}$) and $Q_e$ (mmol g$^{-1}$) represent
the adsorption capacities at a given time $t$ and equilibrium time,
respectively. According to the quantitative analysis, $k_a$ increased
in the order of PEI/$SiO_2$ ($k_a = 0.67$) < 0.15EB-PEI/$SiO_2$
(0.81) < 0.37EB-PEI/$SiO_2$ (0.92) < 0.54EB-PEI/$SiO_2$ (1.2), while
$n_a$ values are all similar in the range of 0.82–0.88. The results
showed that the functionalization of PEI with 1,2-epoxybutane
can significantly enhance the kinetics of $CO_2$ adsorption.

**Long-term stability of the adsorbents.** The long-term stabilities
of the adsorbents were investigated for 50 consecutive TSA
cycles (adsorption: 15% $CO_2$, 3% $H_2O$, 2% Ar in $N_2$ balance at
40 °C; adsorbent regeneration: 100% $CO_2$ at 120 °C). The $CO_2$

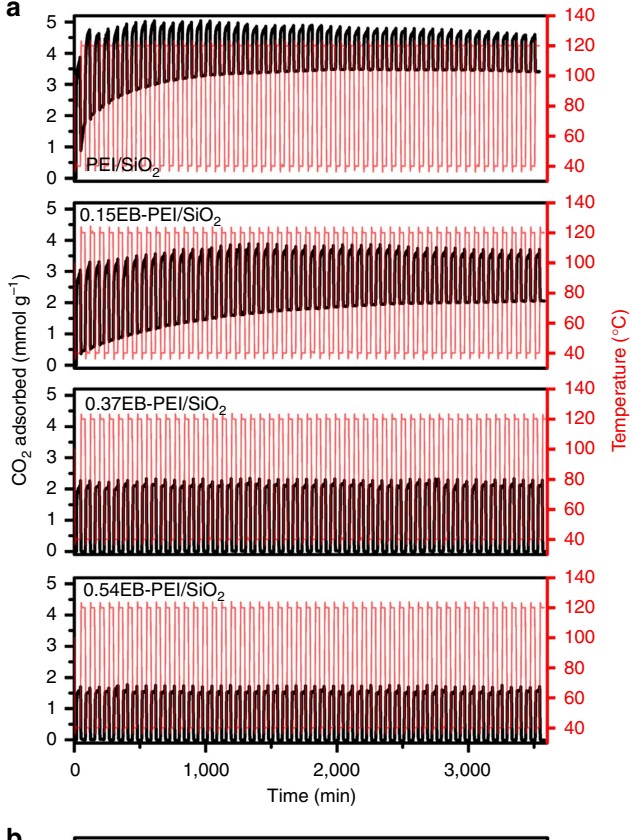

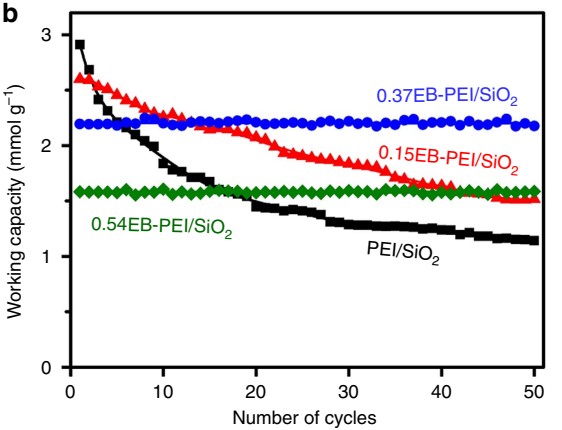

**Figure 4 | Long-term stabilities of the adsorbents.** (**a**) $CO_2$ adsorption–
desorption profiles of the PEI/$SiO_2$, 0.15EB-PEI/$SiO_2$, 0.37EB-PEI/$SiO_2$ and
0.54EB-PEI/$SiO_2$ during 50 consecutive TSA cycles (adsorption: 15% $CO_2$,
3% $H_2O$, 2% Ar in $N_2$ balance at 40 °C; adsorbent regeneration: 100% $CO_2$
at 120 °C). (**b**) $CO_2$ working capacities of adsorbents plotted over the
number of TSA cycles.

adsorption–desorption profiles are shown in Fig. 4a, and the $CO_2$
working capacities at each TSA cycle are summarized in Fig. 4b.
The PEI/$SiO_2$ showed the highest $CO_2$ working capacity
(2.9 mmol g$^{-1}$) in the first cycle, but it rapidly decreased
to 1.1 mmol g$^{-1}$ after 50 TSA cycles. On the other hand,
$n$EB-PEI/$SiO_2$ samples showed more steady cyclic behaviour at
the sacrifice of a part of $CO_2$ working capacity. Such a trend
was more pronounced as the functionalization degree ($n$)
increased. When the functionalization degree was relatively low
(0.15EB-PEI/$SiO_2$), the initial working capacity (2.6 mmol g$^{-1}$)
was only slightly smaller than that of PEI/$SiO_2$. However, the
working capacity still decreased gradually with repeated TSA

cycles, although the decreasing rate became substantially slower than PEI/SiO$_2$. In the case of the optimal functionalization degree (0.37EB-PEI/SiO$_2$), a fairly high CO$_2$ working capacity (2.2 mmol g$^{-1}$) as well as outstanding stability over the repeated TSA cycles was achieved simultaneously. When the functionalization density was increased further (0.54EB-PEI/SiO$_2$), the additional improvement in cyclic stability was only marginal, but the working capacity became unnecessarily small (1.6 mmol g$^{-1}$) due to the large decrease in total amine content (Table 1).

The cyclic stability of adsorbents after the repeated TSA cycles could be correlated with the tendency to form urea. As shown in the Fourier transform infrared (FT-IR) spectra (Fig. 5a), PEI/SiO$_2$ showed highly pronounced peaks for urea species. The infrared spectrum indicated the formation of both open-chain and cyclic urea species, where the formation of cyclic urea is more pronounced. The open-chain urea is formed via dehydrative condensation between CO$_2$ and two amines in different molecules (inter-molecular reaction), while the cyclic urea is formed via

the reaction between CO$_2$ and two amines within a single ethylenediamine unit (intra-molecular reaction; Fig. 5b)[22]. It is notable that $n$EB-PEI/SiO$_2$ samples showed significantly suppressed urea formation, where the degree of suppression increased with a functionalization degree ($n$). Consequently, in the cases of highly functionalized 0.37EB-PEI/SiO$_2$ and 0.54EB-PEI/SiO$_2$ samples, no appreciable formation of urea was detected. The results clearly showed that the functionalization of PEI amine groups with 1,2-epoxybutane remarkably suppressed the urea formation, which in turn improved the stability of the adsorbents over the repeated TSA cycles (Fig. 4a,b).

Sayari and colleagues[23] carried out rigorous mechanistic investigations on urea formation using various amine-functionalized adsorbents[19–23], which can be summarized as shown in Fig. 5b. The formation of open-chain urea most likely involves at least one 1° amine and mainly takes place via an isocyanate pathway (Mechanism A, Route 1). The 1° amine reacts with CO$_2$ to form a carbamic acid, which then dehydrates to form isocyanate. The isocyanate can readily react with either 1° or 2° amine to form a urea. The proposed mechanism was strongly supported by earlier observations by Wu et al.[53] They reported that the reaction between 1° monoamines can form di-substituted ureas, while the mixtures of 1° and 2° monoamines produce di- and tri-substituted ureas. The formation of tetra-substituted urea between 2° monoamines was not observed, which indicates that the urea formation requires at least one 1° amine that can form isocyanate. DFT calculation by Jones and colleagues[24] also supported that the isocyanate pathway is energetically more favoured than the carbamate dehydration pathway (Mechanism A, Route 2). In the case of polyamines (for example, PEI) containing ethylenediamine units (R$_1$NH-CH$_2$-CH$_2$-NHR$_2$), the intra-molecular dehydration to form cyclic ureas (Mechanism B) is favoured[22,54]. Notably, ethylenediamine moieties containing only 2° amines are also known to form cyclic ureas[54], which implies that carbamate dehydration (Mechanism B, Route 2) may also contribute to the formation of a cyclic urea in addition to the energetically favourable isocyanate pathway (Mechanism B, Route 1). It was also reported that steric hindrance near the amine centre can significantly retard the urea formation[54]. On the basis of the earlier discussions, the suppressed urea formation after the functionalization of PEI with 1,2-epoxybutane can be explained by three reasons. First, the functionalization led to the quite selective elimination of 1° amines (Fig. 2b). Considering the fact that isocyanates are produced via the dehydration of carbamic acids derived only from 1° amines, all the energetically favourable isocyanate pathways (Route 1 of both Mechanism A and B) would be significantly suppressed after the functionalization. Second, the 2-ethyl-hydroxyethyl groups (–CH$_2$CH(C$_2$H$_5$)OH) can increase the steric hindrance near amine groups, which can retard the rates of urea formation. Third, the 2-ethyl-hydroxyethyl groups (–CH$_2$CH(C$_2$H$_5$)OH) generated by the functionalization can strongly interact with carbamic acid or carbamate species via hydrogen bonding (solvation effect), which can significantly stabilize them against dehydration reactions.

Finally, the stabilities of the adsorbents against oxidative degradations were also investigated. To evaluate the stabilities in an accelerated manner, the adsorbents were pre-treated in 'CO$_2$-free' synthetic air (20% O$_2$ in N$_2$ balance) at 120 °C for 24 h. It should be noted that the oxidative degradations of amine-containing adsorbents are significantly faster in CO$_2$-free air than in CO$_2$/O$_2$ mixed gases[20]. As shown in the FT-IR spectra measured after the oxidative aging (Supplementary Fig. 9), PEI/SiO$_2$ showed a newly developed infrared peak at 1,680 cm$^{-1}$ consistent with the occurrence of a C=O species. In contrast, as

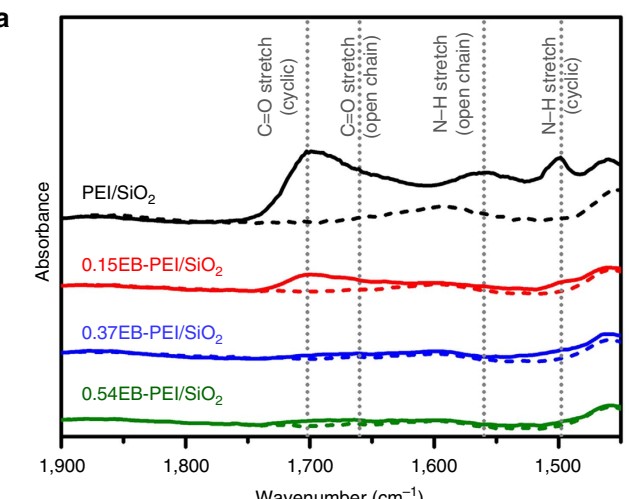

**Figure 5 | Stability against urea formation. (a)** FT-IR spectra of the adsorbents measured after 50 consecutive TSA cycles (adsorption: 15% CO$_2$, 3% H$_2$O, 2% Ar in N$_2$ balance at 40 °C; adsorbent regeneration: 100% CO$_2$ at 120 °C). Dashed lines indicate the spectra for freshly prepared samples, while solid lines indicate those for the samples after 50 consecutive TSA cycles. **(b)** Possible pathways for the CO$_2$-induced urea formation. These mechanisms are based on the previous studies by Sayari and Belmabkhout[21].

the epoxide-functionalization degree ($n$) increased, the evolution of the infrared peak becomes gradually suppressed. In addition, after the oxidative aging, the PEI/SiO$_2$, 0.15EB-PEI/SiO$_2$, 0.37EB-PEI/SiO$_2$ and 0.54EB-PEI/SiO$_2$ showed 19, 56, 80 and 81% remaining CO$_2$ working capacities, respectively, compared with those of the fresh samples (Supplementary Fig. 10). The results clearly showed that the functionalization with 1,2-epoxybutane can also significantly increase resistance against oxidative degradations, which ensures the enhanced stability of the adsorbents in practical operating conditions involving O$_2$. (Note that 3–4% O$_2$ exists in a typical flue gas[2], but O$_2$ concentration will be much lower in the regeneration atmosphere.)

## Discussion

The highly scalable synthesis of a functionalized-PEI/silica composite CO$_2$ adsorbent was demonstrated. The best adsorbent (0.37EB-PEI/SiO$_2$) simultaneously exhibited a large CO$_2$ working capacity (2.2 mmol g$^{-1}$) as well as long-term stability in a practically meaningful TSA process enabling the separation of concentrated CO$_2$ (adsorbent regeneration under 100% CO$_2$ at 120 °C). Most importantly, it was first demonstrated that the functionalization of PEI with 1,2-epoxybutane can significantly enhance CO$_2$ desorption during the adsorbent-regeneration step by reducing the heat of CO$_2$ adsorption. Furthermore, the functionalization resulted in a significantly improved material stability over repeated TSA cycles, due to the marked suppression of urea formation. It was also shown that the functionalization resulted in the significant suppression of the oxidative degradation of amine species, which ensures the long-term stability of the adsorbent in practical operating conditions involving O$_2$. Because the adsorbent synthesis is simple and requires no expensive/corrosive chemicals, it can be readily scaled up. It was also confirmed that 0.37EB-PEI/SiO$_2$ possesses sufficiently high mechanical stability against attrition, which is suitable for fluidized bed operation (Supplementary Note 2). Korea Carbon Capture & Sequestration R&D Center (KCRC) is currently synthesizing the adsorbent with highly reproducible properties at 20 kg scale for test operation in a bench-scale fluidized bed (20 Nm$^3$ h$^{-1}$ scale; Supplementary Figs 11 and 12, Supplementary Table 3 and Supplementary Note 3). It is reasonably expected that various polyamines other than the PEI used in this work can be functionalized with diverse functional epoxides to develop CO$_2$ adsorbents with engineered CO$_2$ adsorption/desorption properties as well as outstanding long-term stability.

## Methods

**Material synthesis.** Functionalization of PEI (Nippon Shokubai, Epomin SP-012, MW 1,200, 22 mmol$_N$ g$^{-1}$) with 1,2-epoxybutane (Sigma-Aldrich, 99%) was carried out by adding varied amounts of 1,2-epoxybutane dropwise into a 17 wt% methanolic solution of PEI. The reaction was carried out at room temperature for 12 h under stirring. Silica microspheres having large porosity were synthesized by spray-drying of a water slurry containing 10 wt% fumed silica (OCI, KONASIL K-300) and 0.5 wt% silica sol (Young Il Chemical, YGS-30) as a binder. In a typical synthesis, 1 kg fumed silica, 0.05 kg silica sol and 8.95 kg water were mixed, and the resultant slurry was injected for spray-drying. The spray-drying was carried out using a spray dryer with a co-current drying configuration and a rotary atomizer (Zeustec ZSD-25, Supplementary Fig. 13). The slurry-feeding rate was 30 cm$^3$ min$^{-1}$, and the rotating speed of atomizer was set to 4,000 r.p.m. The air blowing inlet temperature was 210 °C and the outlet temperature was 150 °C. The resultant silica samples were calcined in dry air at 600 °C to sinter the fumed silica into a 3D porous network. MCM-41 and SBA-15 mesoporous silicas were prepared following the procedures reported previously[47,48]. The polymer–silica composite adsorbents were prepared by wet impregnation of the unmodified PEI and functionalized-PEI methanolic solution (17–28 wt% solution) into the pre-made silica supports. The resultant slurry was dried at 60 °C for 12 h in a vacuum oven to completely remove methanol. The nominal polymer loading was fixed as 50 wt% of the polymer–silica composite adsorbents.

**Material characterization.** Liquid-phase $^{13}$C NMR spectra of the PEI and the functionalized PEIs dissolved in CDCl$_3$ were recorded on an Agilent DD2 400 MHz NMR spectrometer operating at a $^{13}$C frequency of 100.6 MHz. The spectra were obtained with a 1.3 s acquisition time, 25 s relaxation delay, 45° pulse and 256 transients. All $^{13}$C NMR spectra used for quantitative analysis were recorded using inverse-gated proton decoupling to avoid the nuclear Overhauser effect. Chemical shifts were reported in p.p.m. relative to the internal standard of solvents. The nitrogen and carbon elemental contents of the polymers and composite adsorbents were analysed with a FLASH 2000 (Thermo Scientific). The oxygen contents of PEI and the functionalized PEIs were analysed using a FlashEA 112 (Thermo Finnigan). N$_2$ adsorption–desorption isotherms were measured using a Belsorp Max (BEL Japan) volumetric analyser at a liquid N$_2$ temperature ($-196$ °C). Before the measurement, all the samples were degassed at 100 °C for 6 h. The heat of CO$_2$ adsorption of the adsorbents was measured by thermogravimetry-differential scanning calorimetry (Setaram Instrumentation, Setsys Evolution). Before the measurements, the samples were degassed at 100 °C for 1 h under N$_2$ flow (50 cm$^3$ min$^{-1}$). Then, the samples were cooled to 40 °C. Subsequently, the gas was switched to 15% CO$_2$ (50 cm$^3$ min$^{-1}$). The heat of CO$_2$ adsorption was calculated through the integration of the heat-flow curve. For the FT-IR analysis of urea formation, 20 mg of the adsorbents after 50 consecutive TSA cycles were pressed into a self-supporting wafer. Before the FT-IR measurements, each sample was degassed at 100 °C for 6 h under vacuum and cooled to room temperature in an *in situ* infrared cell. FT-IR spectra were collected using an FT-IR spectrometer (Thermo Nicolet NEXUS 470). The FT-IR spectra of freshly prepared samples were additionally measured. For the FT-IR investigation of the adsorbents after oxidative degradation, the adsorbents were similarly pressed into a self-supporting wafer and pre-treated in a synthetic air (20% O$_2$ in N$_2$ balance) at 120 °C for 24 h in the *in situ* infrared cell. After degassing under vacuum, FT-IR spectra were collected at room temperature.

**CO$_2$ adsorption–desorption experiments.** CO$_2$ adsorption–desorption profiles were collected by a TGA-MS setup (Supplementary Fig. 5). Before the measurements, PEI/SiO$_2$ and $n$EB-PEI/SiO$_2$ were degassed at 100 °C for 1 h under N$_2$ flow (50 cm$^3$ min$^{-1}$). CO$_2$ adsorption was carried out using a simulated wet flue gas containing 15% CO$_2$, 3% H$_2$O, 2% Ar (the internal standard for MS calibration) and N$_2$ balance at 40 °C. After 30 min adsorption, the gas was switched to 100% CO$_2$ flow (50 cm$^3$ min$^{-1}$) and the temperature was increased to 120 °C (ramp: 20 °C min$^{-1}$). Then the temperature was maintained for 30 min for the desorption process. The adsorption–desorption cycle was repeated 50 times. The adsorbed amount of CO$_2$ was calculated by subtraction of the adsorbed H$_2$O amount (determined with MS) from the total mass increase determined from TGA. To confirm the reliability of the TGA-MS results, CO$_2$ uptake was also cross-checked with an automated chemisorption analyser (Micromeritics, Autochem II 2920) specially equipped with a cold trap for H$_2$O removal in front of a thermal conductivity detector (Supplementary Fig. 6). In all measurements, samples were diluted 10 times by using sand (quartz) as a diluent for avoiding heat-transfer limitation.

**Data availability.** The data that support the findings of this study are available from the corresponding author on request.

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

## Acknowledgements

This work was supported by the Korea CCS R&D Center (KCRC) grant funded by the Korea government (Ministry of Science, ICT & Future Planning) (NRF-2014M1A8A1049256).

## Author contributions

M.C. designed this study and wrote the manuscript. W.C. and K. M. carried out material synthesis, main material characterizations and $CO_2$ adsorption–desorption experiments. Y.S.K. and J.W.J. carried out FT-IR analysis. H.S. and Y.-K.P. carried out large-scale material synthesis and provided valuable insight into a large-scale $CO_2$ capture process through discussion.

## Additional information

**Competing financial interests:** The authors declare no competing financial interests.

