## [Peer Review File · Nature Communications]

Reviewers' comments:

Reviewer #1 (Remarks to the Author):

This is a very clearly written paper describing a practical synthesis of amine-based sorbents for CO₂ capture from flue gas. The authors have done a thorough job of understanding the prior literature and defining important problems. The goals are framed extremely well, and they are important goals in chemical engineering, materials science, and environmental science.

Comments to address before publication:

1. P2 - "researches" is not used as a noun in English scientific writing - substitute "studies." The plural "researches" is only used as a verb. As a noun, it is always singular.
2. P3 - "in an ambient condition" - change to "at ambient conditions" ; similarly, condition is almost never used in English scientific writing, always the plural - conditions, to describe the state of the experiment, e.g. experimental conditions.
3. P4 - how does fumed silica powder have any porosity? Isn't fumed silica non-porous? Is this porosity in interparticle?
4. P5 - similar to comment 1, behavior, almost never behaviors, plural.
5. P5, 10% error would not be "high reliability" - delete "high"
6. P5, kinetics discussion - while the text suggests that as the epoxide functionalization goes up, kinetics improve, this trend is contradicted somewhat by the data in Fig S5. How are you assessing kinetics to make this claim? Please use a quantitative metric. Visually, the order of approach to equilibrium seems $0.54 > 0.15 > 0.37 > \text{PEI}$.
7. Conclusions - avoid claiming something is economic unless an actual economic analysis is offered in the paper. Claiming scalability seems fine, but I would suggest to always avoid claims of economic viability unless an actual economic analysis is done.
8. Nice figures throughout the paper.
9. Many of the problems addressed in this paper are framed in J. Mater. Chem. 2011, 21, 15100 - it may be worth citing this paper as support for the importance of this work.

Reviewer #2 (Remarks to the Author):

This is a wonderful paper--the authors have simultaneously addressed many of the major challenges for the deployment of solid-supported amines. I very rarely say this, but I believe this manuscript should be published essentially as is. Detailed comments are below:

Summary of the key results

The authors use epoxides to convert primary amines on PEI to secondary amines, and decorate the PEI with a large population of hydroxyl groups. This improves the kinetics by arresting CO₂-induced crosslinking between the primary amines, and improves the cyclic stability of PEI/silica composite by increasing the amount of secondary amines, which are resistant to urea formation. Moreover, they desorb the amines in a CO₂ atmosphere, thus

generating a purified CO₂ product without any difficult process engineering.

Originality and interest: if not novel, please give references

This work is very creative, and shows that small details in amine chemistry have a huge difference in practical implications for utilizing these materials.

Data & methodology: validity of approach, quality of data, quality of presentation

These are all top notch, with the exception of Figure 3a. This figure is somewhat confusing to read, especially since the "2 mmol/g offset" does not actually line up with the "2 mmol/g" tick mark on the y-axis. Is there a better way to present this data? I am aware that the authors are trying to show both capacities and kinetics without the plot being a mess, but the current plot takes a good amount of time to digest and understand.

Appropriate use of statistics and treatment of uncertainties

This could be better--it is not clear how many times the authors repeated these experiments, and I suggest the authors indicate all uncertainties and label their experiments with standard deviations / confidence intervals.

Conclusions: robustness, validity, reliability

The conclusions are well-supported by the data.

Suggested improvements: experiments, data for possible revision

No suggestions other than the ones noted.

References: appropriate credit to previous work?

References are appropriate.

Clarity and context: lucidity of abstract/summary, appropriateness of abstract, introduction and conclusions

The manuscript is very well written and concise.

Reviewer #3 (Remarks to the Author):

Nature Communications covers topics that represent important advances of significance to specialists within each field. In this regard, the current paper MAY fit the scope of Nature Communications; however, the current paper fails to address issues that specialists in the field of high efficiency solid CO₂ sorbents would expect prior to publications. I have

identified the issues that I feel strongly MUST be addressed prior to considering the paper for actual publication. Without dealing with these issues, the current contribution could confuse and mislead less specialized readers. The points that must be clarified and corrected are noted below.

1. The authors cite ref. 10 & 17 regarding the fact that mesopores are not important for designing high-performance CO₂ adsorbents if a silica support possesses sufficiently large 3-dimensional porosity that enables efficient CO₂ diffusion. They focus on formation of silica microspheres having extra-large porosity using low cost fumed silica, which they report creates polymer-silica composite CO₂ adsorbents via solutions of PEI and nEB-PEIs were impregnated into the pre-made porous silica microspheres. The authors show porous silica microspheres with sizes of 80 - 120 micrometers, Fig. 2c, and which had significantly larger pore volume (2.1 cm³ g⁻¹) than the original fumed silica powder (300 m² g⁻¹ and 1.0 cm³ g⁻¹) (Table 1 and Fig. 2e). They also note that these microspheres are easily produced in kg-scale. In the supplemental material (Fig. S8) they show a large spray dryer used for silica synthesis. Clearly, they must have ACTUAL COMPLETE PARTICLE SIZE DISTRIBUTIONS that support the claim of the 90-120 micrometer sizes. Such complete particle size distributions should be included in the supplemental material, since it is somewhat surprising to me that such a tight distribution could be achieved.

2. Related to the above point, the large pore volume (2.1 cm³ g⁻¹), which is more than twice that of the original fumed silica powder (300 m² g⁻¹ and 1.0 cm³ g⁻¹) suggests that the particles would be "fluffy" and difficult to handle. The authors note on page 4 that Korea Carbon Capture & Sequestration R&D Center (KCRC) is currently synthesizing the adsorbent with highly reproducible properties at 20 kg scale for test operation in a bench-scale fluidized bed (20 Nm³ h⁻¹ scale). This plan suggest that the material can be handled adequately; however, the details of how the noted bench scale fluidized bed is to operate should be included in the Supplemental material to provide credibility of possible viability of the work with these low density particles.

3. The testing of sorption and desorption to simulate a TSA cycle is described in what seems to be a TGA system in Fig. S3. The breakthrough tests were done with a small fixed bed set up roughly shown in Fig. S4; however, the information provided is greatly inadequate. With the large CO₂ capacity claimed for the sorbents here, even with less intense enthalpies of sorption shown in Fig. 1, a considerable heat must be removed during sorption to avoid problems. These are not small details, and readers should be able to understand and reproduce the work by know how such temperature excursions were avoided to enable the lack of apparent heat transfer limitations. In fact, perhaps this issue may explain the greatly protracted sorption kinetics in Fig. 3.

4. While such heat transfer issues may be avoided in a fluidized bed, it is NOT CLEAR THAT THE VERY PROTRACTED KINETICS ARE ONLY DUE TO THESE EFFECTS. Other researchers have found that when pores in a solid supported amine are highly loaded with amines, as the authors have done here, the polymer backbones become rigidified and GREATLY REDUCE THE DIFFUSION COEFFICIENTS WITHIN THE PARTICLE. If this effect is at play, the sorption AND desorption processes within the rather large particle will probably show greatly

reduced mass transfer and greatly hamper processes EVEN IF THE ABOVE-MENTIONED THERMAL PROBLEMS IN A FIXED BED ARE MITIGATED IN A FLUIDIZED BED. This issue can be simply addressed by decoupling these thermal issues from the intrinsic mass transfer issues by diluting the sorbent particles in a TGA test with an inert component such as sand that has negligible sorption but provides thermal moderation through its heat capacity. If the protracted kinetics are still seen (WHICH I EXPECT TO BE THE CASE) this will be indicative of a serious mass transfer limitation due to reduced diffusion coefficients in the CO₂ loaded amine polymer in the pores.

5. Finally, the extremely low bulk density of the particles would seem to make the particles rather fragile and not suitable for an actual fluidized bed process in which they are proposed to be used. I think the authors MUST have done some work to characterize the handling and attrition tendency of the particles that they have produced, and this MUST be discussed and characterized before the work is considered seriously for publication. I also expect that swelling stresses and even hydrolysis of the sorbents may be a serious issue. The authors note that they have exposed the sorbents to 50 TGA cycles. Based on Fig. 4, it appears that this comprises less than 100 hours of exposure in the TGA mode to simulate TSA cycles. I think this issue may be serious, since during the high temperature regeneration, water will be present along with the sorbed CO₂, and this may cause long term degradation of the sorbents.

6. Despite the above concerns, the optimal functionalization degree (0.37EB-PEI/SiO₂) shows an attractive CO₂ working capacity (2.2 mmol g⁻¹) and good stability over the repeated simulated TSA cycles, so the work would be appropriate for publication if the authors address the 5 preceding concerns properly.

Point-by-Point Response

Reviewer #1

This is a very clearly written paper describing a practical synthesis of amine-based sorbents for CO₂ capture from flue gas. The authors have done a thorough job of understanding the prior literature and defining important problems. The goals are framed extremely well, and they are important goals in chemical engineering, materials science, and environmental science.

Point 1: "researches" is not used as a noun in English scientific writing - substitute "studies." The plural "researches" is only used as a verb. As a noun, it is always singular.

Response 1: We greatly appreciate the English correction. The manuscript was corrected accordingly.

Point 2: "in an ambient condition" - change to "at ambient conditions"; similarly, condition is almost never used in English scientific writing, always the plural - conditions, to describe the state of the experiment, e.g. experimental conditions.

Response 2: The manuscript was corrected accordingly.

Point 3: how does fumed silica powder have any porosity? Isn't fumed silica non-porous? Is this porosity in interparticle?

Response 3: Yes. It is an inter-particle porosity. We clarified this point on page 4, line 21 of the revised manuscript.

Point 4: Similar to comment 1, behavior, almost never behaviors, plural.

Response 4: The manuscript was corrected accordingly.

Point 5: 10% error would not be "high reliability" - delete "high"

Response 5: We agree with the reviewer's suggestion. The manuscript was corrected accordingly.

Point 6: kinetics discussion - while the text suggests that as the epoxide functionalization goes up, kinetics improve, this trend is contradicted somewhat by the data in Fig S5. How are you assessing kinetics to make this claim? Please use a quantitative metric. Visually, the order of approach to equilibrium seems 0.54>0.15>0.37>PEI.

Response 6: We greatly appreciate the reviewer's important comments. The CO₂ uptake curves in Figure S5 (original version) were measured with a setup that was built by modifying a commercial TPR instrument (Micromeritics, Autochem II 2920) equipped with a TCD detector. We used this instrument to cross-check the equilibrium adsorption capacities measured with our main instrument, a TGA-MS setup. The measurements were

carried out by placing small amounts of adsorbent samples at the bottom of a conventional U-tube TPR cell. This is not a carefully prepared packed bed and cannot give us reliable kinetic information. Furthermore, as reviewer 3 has pointed out (Point 14 and 15), we were not rigorous in considering the heat transfer limitation during the CO₂ adsorption/desorption experiments.

In the revised manuscript, we completely re-measured the data by carefully preparing a packed bed in the vertical region of a U-tube cell. To obtain reliable kinetics data, samples were diluted 10 times by using sand (quartz) as a diluent in order to avoid the heat transfer limitation. For quantitative analysis of the CO₂ adsorption kinetics, CO₂ uptake profiles measured with the breakthrough experimental setup were fitted with the Avrami kinetic equation [Wang, X., Chen, L. & Guo, Q. Development of hybrid amine-functionalized MCM-41 sorbents for CO₂ capture. *Chem. Eng. J.* **260**, 573–581 (2015)]:

$$Q_t = Q_e[1 - \exp((-k_a t)^{n_a})]$$

, where k_a (min⁻¹) is a rate constant, n_a is the kinetic order of the Avrami model, and Q_t (mmol g⁻¹) and Q_e (mmol g⁻¹) represent the adsorption capacities at a given time t and equilibrium time, respectively. In Figure S8 (shown below), the solid lines indicate the CO₂ uptake profiles of the adsorbents measured with the breakthrough experimental setup, and the dotted lines indicate the fitting curves obtained with the Avrami kinetic equation. According to the quantitative analysis, k_a increased in the order of PEI/SiO₂ ($k_a = 0.67$) < 0.15EB-PEI/SiO₂ (0.81) < 0.37EB-PEI/SiO₂ (0.92) < 0.54EB-PEI/SiO₂ (1.2) while n_a values are all similar in the range of 0.82 – 0.88 (see Table S2 for fitting results). The results showed that the functionalization of PEI with 1,2-epoxybutane can significantly enhance the kinetics of CO₂ adsorption.

Figure S8 | CO₂ adsorption profiles of the adsorbents measured with the breakthrough experimental setup.

CO₂ adsorption was carried out in a wet flue gas containing 15% CO₂, 3% H₂O and N₂ balance at 40 °C. The solid lines indicate the CO₂ uptake profiles of the adsorbents measured experimentally, and the dotted lines indicate the fitting curves obtained with the Avrami kinetic equation.

Table S2 | Kinetic fitting results of the CO₂ adsorption profiles (measured with a breakthrough setup) using Avrami equation.

parameters	PEI/SiO ₂	0.15EB-PEI/SiO ₂	0.37EB-PEI/SiO ₂	0.54EB-PEI/SiO ₂
k_a	0.67	0.81	0.92	1.2
n_a	0.88	0.83	0.82	0.82
R^2	0.9876	0.9705	0.9530	0.9507

Point 7: Conclusions - avoid claiming something is economic unless an actual economic analysis is offered in the paper. Claiming scalability seems fine, but I would suggest to always avoid claims of economic viability unless an actual economic analysis is done.

Response 7: We fully agree with the reviewer's thoughtful comments. We removed 'economic' from the manuscript.

Point 8: Nice figures throughout the paper.

Response 8: We greatly appreciate the kind words.

Point 9: Many of the problems addressed in this paper are framed in J. Mater. Chem. 2011, 21, 15100 - it may be worth citing this paper as support for the importance of this work.

Response 9: The suggested reference is a review paper that is highly relevant to our discussion. We added the reference in the revised manuscript. We greatly appreciate the kind suggestion.

Reviewer #2

This is a wonderful paper--the authors have simultaneously addressed many of the major challenges for the deployment of solid-supported amines. I very rarely say this, but I believe this manuscript should be published essentially as is. Detailed comments are below:

The authors use epoxides to convert primary amines on PEI to secondary amines, and decorate the PEI with a large population of hydroxyl groups. This improves the kinetics by arresting CO₂-induced crosslinking between the primary amines, and improves the cyclic stability of PEI/silica composite by increasing the amount of secondary amines, which are resistant to urea formation. Moreover, they desorb the amines in a CO₂ atmosphere, thus generating a purified CO₂ product without any difficult process engineering. This work is very creative, and shows that small details in amine chemistry have a huge difference in practical implications for utilizing these materials.

Point 10: These are all top notch, with the exception of Figure 3a. This figure is somewhat confusing to read, especially since the "2 mmol/g offset" does not actually line up with the "2 mmol/g" tick mark on the y-axis. Is there a better way to present this data? I am aware that the authors are trying to show both capacities and kinetics without the plot being a mess, but the current plot takes a good amount of time to digest and understand.

Response 10: We appreciate the reviewer's thoughtful suggestion. To enhance readability, we changed the scale of the temperature profile. In the revised Figure 3a, "2 mmol g⁻¹ offset" lines up with the "2 mmol g⁻¹" tick mark on the y-axis.

Reviewer #3

Point 11: Nature Communications covers topics that represent important advances of significance to specialists within each field. In this regard, the current paper MAY fit the scope of Nature Communications; however, the current paper fails to address issues that specialists in the field of high efficiency solid CO₂ sorbents would expect prior to publications. I have identified the issues that I feel strongly MUST be addressed prior to considering the paper for actual publication. Without dealing with these issues, the current contribution could confuse and mislead less specialized readers. The points that must be clarified and corrected are noted below.

Response 11: We greatly appreciate the reviewer's constructive comments. The reviewer's comments attached below were truly helpful in improving the quality and rigorousness of the present manuscript.

Point 12: The authors cite ref. 10 & 17 regarding the fact that mesopores are not important for designing high-performance CO₂ adsorbents if a silica support possesses sufficiently large 3-dimensional porosity that enables efficient CO₂ diffusion. They focus on formation of silica microspheres having extra-large porosity using low cost fumed silica, which they report creates polymer-silica composite CO₂ adsorbents via solutions of PEI and *n*EB-PEIs were impregnated into the pre-made porous silica microspheres. The authors show porous silica microspheres with sizes of 80 - 120 micrometers, Fig. 2c, and which had significantly larger pore volume (2.1 cm³ g⁻¹) than the original fumed silica powder (300 m² g⁻¹ and 1.0 cm³ g⁻¹) (Table 1 and Fig. 2e). They also note that these microspheres are easily produced in kg-scale. In the supplemental material (Fig. S8) they show a large spray dryer used for silica synthesis. Clearly, they must have ACTUAL COMPLETE PARTICLE SIZE DISTRIBUTIONS that support the claim of the 80-120 micrometer sizes. Such complete particle size distributions should be included in the supplemental material, since it is somewhat surprising to me that such a tight distribution could be achieved.

Response 12: We greatly appreciate the reviewer's thoughtful comments. The original statement was made based on a rather crude analysis of SEM images. Following the reviewer's suggestion, we added an actual complete particle size distribution measured using a particle size analyzer (Figure S2, also shown below). The complete particle distribution is broader than the size distribution we originally stated. >90% of the silica particles have a diameter in the range of 75 – 200 micrometers. Therefore, we wrote on page 4, line 16 – 19 of the revised manuscript: “The silica spheres have particle sizes mainly in the range of 75 – 200 micrometers (> 90 %, the complete particle size distribution is given in Figure S2), which is suitable for fluidized bed operations in a large-scale CO₂ capture process.”

Figure S2 | Particle size distribution of the silica spheres. The particle size distribution was measured using Microtrac Bluewave particle size analyser.

Point 13: Related to the above point, the large pore volume ($2.1 \text{ cm}^3 \text{ g}^{-1}$), which is more than twice that of the original fumed silica powder ($300 \text{ m}^2 \text{ g}^{-1}$ and $1.0 \text{ cm}^3 \text{ g}^{-1}$) suggests that the particles would be "fluffy" and difficult to handle. The authors note on page 4 that Korea Carbon Capture & Sequestration R&D Center (KCRC) is currently synthesizing the adsorbent with highly reproducible properties at 20 kg scale for test operation in a bench-scale fluidized bed ($20 \text{ Nm}^3 \text{ h}^{-1}$ scale). This plan suggest that the material can be handled adequately; however, the details of how the noted bench scale fluidized bed is to operate should be included in the Supplemental material to provide credibility of possible viability of the work with these low density particles.

Response 13: As the reviewer pointed out, the pure silica spheres are rather fluffy, but polyethyleneimine-impregnated silica spheres are much less so. Please note that we filled the major portion of the silica pores with polyethyleneimine, and there was quite small residual porosity. Of course, compared with purely inorganic materials such as zeolite, the density of the present composite adsorbent is somewhat lower (apparent density: 0.6). Nevertheless, the density was sufficiently high for successful fluidization. Please note that RTI International has already succeeded in the fluidized operation of PEI/SiO₂ adsorbents. According to their report [Nelson, T. O. *et al.* Advanced solid sorbent-based CO₂ capture process. *Energy Procedia* **63**, 2216–2229 (2014)], the apparent density of their adsorbent (0.6 g mL^{-1}) is also similar to ours.

In the revised Supplementary Information, we also added details of our temporary bench-scale fluidized bed operation in KCRC (Figures S11–12 and Table S3, also shown below). Our existing fluidized bed system is composed of 3 sets of carbonator and regenerator. The multi-purpose facility was specially designed to use the adsorption energy released in one carbonator for the regeneration of adsorbents in the next regenerator. It was originally designed to combine three kinds of adsorbents whose adsorption and regeneration temperatures are different. That system was already disclosed at GHGT 12 conference in 2014 [Park, Y. K. *et al.* Energy recoverable multi-stage dry sorbent CO₂ capture process. *Energy Procedia* **63**, 2266–2279 (2014)]. All the adsorbents and regenerators are riser-type. Adsorbents are introduced from the bottom and fluidized by flue gases and sweeping

gases. The actual facility is made of four 4 m-tall core-shell type cylinders with 6 cyclones. The volume of the core of a cylinder is 15.5 L, and the volume of the shell is 3.4 L. The core of the first cylinder is the adsorber of the low-temperature stage. The shell of the first cylinder is where cooling water flows to remove the heat of adsorption from the low-temperature adsorbents. The shell of the second cylinder is the desorber of the low-temperature stage, and its core is the adsorber of the medium-temperature stage. The shell of the third cylinder is the desorber of the medium-temperature stage, and the core of the third cylinder is the adsorber of the high-temperature stage. The shell of the fourth cylinder is the adsorber of the high-temperature stage, and an electric heater is in the core of the fourth cylinder. The regeneration energy is supplied only to the regenerator of the high-temperature stage, and the regeneration energy for the medium-temperature stage is supplied from the adsorption energy of the high temperature stage. The regeneration heat of the low-temperature stage is supplied from the medium-temperature stage.

Figure S11 | Fluidized bed setup used for test operation in a bench-scale. a-b, Schematic (a), and photograph (b) of the fluidized bed.

Our adsorbents were temporarily tested in the low-temperature stage of this facility (KCRC is currently building a new bench-scale fluidized bed facility which will be fully dedicated to the long-term test of the present adsorbents). The energy needed to run the process was supplied from the electric heater by heat transfer between spherical zeolite particles circulating in the medium-temperature stage and the high-temperature stage. A simulated flue gas containing 12% CO₂ and 3% H₂O was fed into the adsorber from the bottom of the adsorber blowing the adsorbents entering the adsorber from the connected standpipe. The solid circulation rate was controlled by regulating the pressure difference of the standpipe of the adsorber. After being separated from the CO₂-depleted flue gas by a cyclone, the carbonated adsorbents were accumulated in the standpipe of the regenerator and fed to the regenerator with sweep gas. After being separated from the product gas by a cyclone, the adsorbents were accumulated in the standpipe of the adsorber and fed to the adsorber again. Table S3 of the revised supplementary information demonstrates the stabilized operating condition of the facility, and Figure S12 shows results for the stabilized operation of the facility for 2 h. The adsorption temperature was measured at the

end of the adsorber, and the desorption temperature was measured at the end of the regenerator. About 80% of the CO₂ from the simulated flue gas was captured. The pressure differences and the temperatures of the adsorber and the regenerator were constantly maintained.

Figure S12 | Results of bench-scale CO₂ adsorption experiments conducted in the low-temperature stage of the 3-stage fluidized bed facility.

Table S3 | Stabilized operating condition in the fluidized bed facility.

Parameters	Values
Feed gas volume flow (L min ⁻¹)	175
Sweep gas volume flow (L min ⁻¹)	15
Solid circulation rate (kg h ⁻¹)	55
Feed gas mole fraction	CO ₂ (0.120), H ₂ O (0.030), O ₂ (0.187), N ₂ (0.663)
Sweep gas mole fraction	CO ₂ (1)
Adsorption temperature (°C)	50
Desorption temperature (°C)	150

Point 14: The testing of sorption and desorption to simulate a TSA cycle is described in what seems to be a TGA system in Fig. S3. The breakthrough tests were done with a small fixed bed set up roughly shown in Fig. S4;

however, the information provided is greatly inadequate. With the large CO₂ capacity claimed for the sorbents here, even with less intense enthalpies of sorption shown in Fig. 1, a considerable heat must be removed during sorption to avoid problems. These are not small details, and readers should be able to understand and reproduce the work by know how such temperature excursions were avoided to enable the lack of apparent heat transfer limitations. In fact, perhaps this issue may explain the greatly protracted sorption kinetics in Fig. 3.

Response 14: We greatly appreciate the reviewer`s very thoughtful and important comments. We fully agree that heat transfer is not a small detail, especially in analyzing kinetics (equilibrium capacities would not be affected). Following the reviewer`s critical suggestion, we diluted our adsorbents with sand (quartz) in different ratios and studied its effect on the kinetics of CO₂ adsorption in both the TGA-MS (Figure S7a) and breakthrough setups (Figure S7b).

Figure S7 | CO₂ adsorption profiles of the PEI/SiO₂ after dilution with sand in different ratios. a-b, CO₂ adsorption profiles in TGA-MS (a) and breakthrough setup (b).

The results show that CO₂ adsorption kinetics becomes substantially enhanced in both systems, as the adsorbent is diluted with more sand. Above 10-fold dilution, however, further enhancement was only marginal, which means that this dilution ratio is sufficient for avoiding heat transfer limitation. Based on these data, we completely re-measured all the CO₂ adsorption-desorption profiles (Figure 2f, Figure 3a, Figure 4a, Figure S8, and Figure S10 of the revised manuscript) after 10-fold dilution of the adsorbents with sand.

Point 15: While such heat transfer issues may be avoided in a fluidized bed, it is NOT CLEAR THAT THE VERY PROTRACTED KINETICS ARE ONLY DUE TO THESE EFFECTS. Other researchers have found that when pores in a solid supported amine are highly loaded with amines, as the authors have done here, the polymer backbones become rigidified and GREATLY REDUCE THE DIFFUSION COEFFICIENTS WITHIN THE PARTICLE. If this effect is at play, the sorption AND desorption processes within the rather large particle will probably show greatly reduced mass transfer and greatly hamper processes EVEN IF THE ABOVE-MENTIONED THERMAL PROBLEMS IN A FIXED BED ARE MITIGATED IN A FLUIDIZED BED. This issue can be simply addressed by decoupling there thermal issues from the intrinsic mass transfer issues by diluting

the sorbent particles in a TGA test with an inert component such as sand that has negligible sorption but provides thermal moderation through its heat capacity. If the protracted kinetics are still seen (WHICH I EXPECT TO BE THE CASE) this will be indicative a serious mass transfer limitation due to reduced diffusion coefficients in the CO₂ loaded amine polymer in the pores.

Response 15: As explained in Response 14, the adsorption kinetics were significantly enhanced after the 10-fold dilution with sand. The results indicate that the very protracted kinetics in our earlier measurements were mainly due to heat transfer limitation. Please also note that the kinetic analysis of the newly measured data (Figure S8) showed that the epoxide functionalization of PEI can significantly increase the CO₂ adsorption kinetics (see Response 6).

Point 16: Finally, the extremely low bulk density of the particles would seem to make the particles rather fragile and not suitable for an actual fluidized bed process in which they are proposed to be used. I think the authors MUST have done some work to characterize the handling and attrition tendency of the particles that they have produced, and this MUST be discussed and characterized before the work is considered seriously for publication. I also expect that swelling stresses and even hydrolysis of the sorbents may be a serious issue. The authors note that they have exposed the sorbents to 50 TGA cycles. Based on Fig. 4, it appears that this comprises less than 100 hours of exposure in the TGA mode to simulate TSA cycles. I think this issue may be serious, since during the high temperature regeneration, water will be present along with the sorbed CO₂, and this may cause long term degradation of the sorbents.

Response 16: This is a thoughtful and important comment. In fact, a small amount of binder (5 wt% silica sol) was added to the fumed silica slurry before spray drying, in order to achieve sufficient mechanical stability required for fluidized bed operation. In the original manuscript, we focused on the epoxide functionalization of PEI and did not carefully mention this point. The addition of a binder, of course, results in the enhancement of the mechanical strength of the silica bead at the sacrifice of some porosity ($2.1 \rightarrow 1.7 \text{ cm}^3 \text{ g}^{-1}$). To address the reviewer's comments, we completely re-measured all the data (pore structural data and the CO₂ adsorption-desorption profiles of the composite adsorbents) by using the actual silica microspheres we tested for our fluidized operation (*i.e.*, silica produced using 5 wt% silica sol as a binder). Because we did not carefully consider heat transfer limitation in the original experiments (Points 14 and 15), all the CO₂ adsorption-desorption profiles should have been re-measured. Therefore, we completely changed our sample system and collected all the data again (this is why we spent fully 2 months for the revision). In the 'Methods' section of the revised manuscript, we also clearly stated that a silica sol (Young Il Chemical, YGS-30) was added to the fumed silica slurry as a binder.

As the reviewer suggested, the attrition behavior of the silica microspheres and 0.37EB-PEI/SiO₂ samples were measured using a three-hole airjet attrition tester configured following ASTM D5757-95. 50 g of adsorbent were used for each test. As specified in the ASTM method, the test was carried out under 10 L min⁻¹ air flow and the weight loss of fines was recorded after 5 h time-on-stream. The percentage of fines of pure silica microspheres collected in a thimble was 74%, while that of 0.37EB-PEI/SiO₂ was only 2.5%. The results clearly showed that the impregnation of organic polymer dramatically increased the mechanical stability of the silica

microspheres. The results were discussed in Note 3 of the Supplementary Information.

The long-term steam stabilities of the composite adsorbents made from MCM-41, SBA-15, and our silica microspheres were also compared. We treated the PEI/silica composite adsorbents under 100% steam at 120 °C for 7 days. After the steam treatment, the composite adsorbents were calcined at 600 °C and the pore structures of the silica supports were analyzed using N₂ adsorption-desorption at -196 °C (Figure S4 of the revised supplementary information, also shown below). The results showed that the N₂ adsorption-desorption isotherm of the steam-treated silica microsphere changed only slightly (<5% pore volume loss) compared with that of a pristine silica microsphere. In clear contrast, the isotherms of SBA-15 and MCM-41 dramatically changed after the steam treatment compared to those of pristine silicas. 39% and 46% decreases in pore volume were observed for SBA-15 and MCM-41, respectively. The present results clearly indicate that the present silica microspheres possess much higher hydrothermal stability compared with ordered mesoporous silicas such as SBA-15 and MCM-41. The remarkably higher stability of the silica microspheres can be attributed to their significantly larger silica framework thickness (10 – 15 nm) than those of SBA-15 (ca. 3 nm) and MCM-41 (ca. 1 nm). The present results were discussed on page 4, line 29 of the revised manuscript.

Figure S4 | Steam-stability of PEI/silica adsorbents. a-c, N₂ adsorption-desorption isotherms of PEI-impregnated silica microspheres (PEI/SiO₂) (a), PEI/SBA-15 (b), and PEI/MCM-41 (c) after treatment under 100% steam at 120 °C for 7 days, followed by calcination at 600 °C.

Point 17: Despite the above concerns, the optimal functionalization degree (0.37EB-PEI/SiO₂) shows an attractive CO₂ working capacity (2.2 mmol g⁻¹) and good stability over the repeated simulated TSA cycles, so the work would be appropriate for publication if the authors address the 5 preceding concerns properly.

Response 17: We greatly appreciate the aforementioned reviewer's important comments. The additional experiments suggested by all the reviewers were truly helpful in improving the quality of data and in producing a clearer interpretation. We believe that we have now fully addressed all the reviewer's comments.

REVIEWERS' COMMENTS:

Reviewer #3 (Remarks to the Author):

The authors have done a careful job of responding to each of my 5 concerns.

It was impressive to see the detailed follow-up to address the technical items that I was concerned about, and I think the paper can be published as it is, since they have carefully indicated what they did.

The response to my 3rd and 4th points requesting sand dilution studies (which they referred to as points 14 & 15 in their response) clarifies the magnitude of the mass transfer vs. heat transfer limitations. By removing the heat transfer issues, the authors have now clearly shown that magnitude of the improvement with their EB-PEI/SiO₂ samples relative to PEI/SiO₂, and the results are worth reporting.

The response to my point 5 (which they refer to as point 16) using both the ASTM test and steam stability testing is also impressive--and surprising. In any case, they have done the work to address all of the above issues, so I am satisfied that the work can be published.